# 3D histology of human heart-forming organoids by X-ray phase-contrast tomography

Karlo Komorowski [1] ✉, Jakob Reichmann [2], Lika Drakhlis [3], Robert Zweigerdt [3] & Tim Salditt [2]

Three-dimensional (3D) imaging is crucial for elucidating the complex structure of organoid models which involve complex spatial cellular and tissue organization in 3D. While a variety of volume imaging methods, including novel light microscopy tools, are now well established to probe the cellular complexity of organoids in 3D, the gold standard for obtaining a precise morphological picture is histology, a traditionally 2D imaging technique that relies on slicing the specimen and therefore has severe limitations in scalability and volumetric imaging. X-ray phase-contrast tomography (XPCT) has emerged as an imaging modality capable of extending conventional histology into the third dimension. While it has been applied to various types of animal and human tissues, its applicability to organoid systems, however, is yet in its infancy. Here, we use XPCT for 3D histology of unstained and formalin-fixed paraffin-embedded human heart-forming organoids (HFOs) at multiple scales and with isotropic resolution. Derived from human pluripotent stem cells, HFOs are a complex and highly structured in vitro model of early heart, foregut and vasculature development, resembling the early human heart-forming region. Using highly coherent synchrotron radiation, we show that HFOs and their different tissue elements can be visualized in their full three-dimensionality and at subcellular scale.

Organoids are near-physiological three-dimensional (3D) in vitro models, which are derived from stem or progenitor cells, and can be regarded as simplified representations of an organ. They reflect key spatial structure and organ functionality of their in vivo counterparts[1,2]. In this still rapidly developing field of 'mini-organs in a dish', they have proven to be valuable tools for a wide range of applications in basic and translational research, e.g., in developmental biology, regenerative medicine, and pharmacological or cancer research. The self-organization from (3D) stem or progenitor cell cultures, induced by physical and biochemical cues, results in highly complex 3D cultures potentially with different connected tissue patterns. The complex 3D morphology, in turn, can be modulated or controlled by manipulation technologies such as gene-targeting, for example in disease modeling, or drug-testing[3–5].

Imaging is of central importance to probe the complex morphology of organoids. In particular, volume imaging methods are required to fully capture the micro-anatomy and cellular complexity of the 3D structure. The development of organoid systems over the past decade has greatly benefited from microscopic approaches such as confocal, multiphoton, and more recently light sheet fluorescence microscopy, to probe the 3D tissue structure together with detailed molecular information, i.e., by visualizing single-cell gene expression, as reviewed in refs. [6–8]. These fluorescence-based techniques, however, require extensive and time-consuming sample preparation and the morphological information content is limited to the labeled tissue components. Morphological information is typically assessed by optical microscopy of thin histological sections either stained with specific staining agents or in combination with immunohistochemistry. While histology provides very good results in 2D, an extension to 3D imaging is possible in principle in terms of serial sections, but the resolution is generally limited by the slice thickness in the third dimension.

The unique potential of X-ray phase-contrast tomography (XPCT) to extend histology into the third dimension has been demonstrated in recent years for tissues from various organs, such as cardiac tissue[9,10], lung tissue[11,12], and brain tissue[13–15], both at the synchrotron and at compact laboratory sources[16–18]. Contrast in soft tissue can be achieved without staining the specimen by exploiting X-ray phase shifts in the tissue, rather than absorption as in conventional computed tomography (CT). The phase

[1]Department of Radiation Protection and Medical Physics, Hannover Medical School, Hannover, Germany. [2]Institute for X-Ray Physics, University of Göttingen, Göttingen, Germany. [3]Leibniz Research Laboratories for Biotechnology and Artificial Organs (LEBAO), Department of Cardiothoracic, Transplantation and Vascular Surgery (HTTG), REBIRTH-Research Center for Translational Regenerative Medicine, Hannover Medical School, Hannover, Germany. ✉e-mail: Komorowski.Karlo@mh-hannover.de

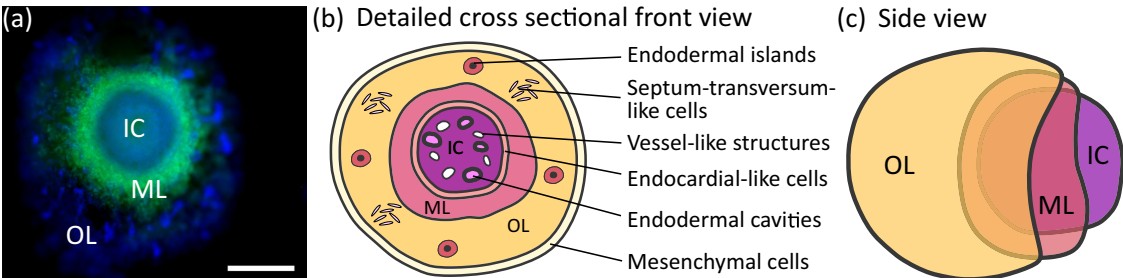

**Fig. 1 | The human heart-forming organoid. a** A microscopic image of a typical HES3 NKX2.5-eGFP-derived HFO (green: NKX2.5-eGFP, blue: DAPI) shows the characteristic layers IC, ML, and OL, scalebar is 500 μm. **b** A detailed sketch of a cross sectional front view shows different characteristic features of the organoid as indicated. **c** A side view of an HFO schematically shows the distinct layers IC, ML, and OL. The sketches in (**b**, **c**) are adapted from ref. 21.

shifts can be measured in so-called phase-contrast CT, and result from the spatial variation of the real-valued part $\delta(\mathbf{r})$ of the X-ray index of refraction $n(\mathbf{r}) = 1 - \delta(\mathbf{r}) - i\beta(\mathbf{r})$, which in soft tissue is orders of magnitude larger than the imaginary part $\beta(\mathbf{r})$. Note that $\beta$ is directly proportional to the attenuation coefficient which is exploited in conventional X-ray imaging. In free-space propagation, the phase shifts are transformed into measurable intensity variations by self-interference of a partially coherent beam behind the sample[19]. Compared to other phase-sensitive X-ray imaging techniques, e.g., based on grating interferometry[20], propagation-based XPCT does not rely on additional X-ray optics between the sample and the detector and stands out due to its high dose efficiency and the capability of higher spatial resolution. Importantly, XPCT offers a full 3D visualization with scalable and isotropic resolution without destructive slicing of the specimen and quantitative contrast. Note, that the refractive decrement $\delta(\mathbf{r})$ is proportional to the local electron density.

In this work, we demonstrate a multiscale approach for 3D virtual histology of recently established human heart-forming organoids (HFOs)[21,22] based on XPCT. HFOs, derived from human pluripotent stem cells (hPSCs), including human embryonic stem cells (hESCs) and induced pluripotent stem cells (iPSCs)), are the first in vitro model properly recapitulating key properties of early native heart, foregut, and vasculature development. These organoids show a characteristic layered pattern consisting of an inner core (IC), a myocardial layer (ML), and an outer layer (OL), see Fig. 1. The entire organoid, but particularly the IC, is permeated by a vascular endothelial cell network. Importantly, HFOs resemble the early human heart-forming region that develops in close proximity to the foregut endoderm. Spatially and morphologically distinct foregut endoderm tissues are located in the IC and OL, anterior foregut endoderm (AFE) and posterior foregut endoderm (PFE) respectively, giving rise to lung, esophagus, and other organs in the embryo. In addition, HFOs have been used as a model system to study genetic defects by demonstrating that *NKX2.5* knockout (KO) induces malformations in the ML similar to defects in heart development observed in transgenic mice[21].

To study the complex 3D morphology of these highly structured multi-tissue organoids, we use two different X-ray optical configurations for synchrotron radiation, namely the parallel-beam (SR-PB) setup and the cone-beam (SR-CB) setup, both implemented at the Göttingen Instrument for Nano-Imaging with X-Rays (GINIX) endstation of the P10 beamline at the Deutsches Elektronensynchrotron (DESY, Hamburg, Germany), and in addition a commercial laboratory μ-CT equipped with a nano-focus source, schematically shown in Fig. 2. The SR-PB setup offers a large field-of-view (FOV) of $1.5 \times 1.5$ mm$^2$ with a pixel size of 650 nm, limited by the pixel size of the detector, and a high-throughput mode by continuous scan and detector readout, which enables to scan the entire biopsy punch within minutes. In the SR-CB geometry, selected sub-regions can be scanned with effective pixel sizes ranging from 50 to 300 nm, depending on the geometric magnification.

In this manuscript, we first describe details of sample preparation, tomographic instrumentation, and data acquisition, post-processing of tomographic data, and image analysis in the Methods section. In the following Results section, we demonstrate that important structural features of interest including the different tissue layers of an HFO can be effectively visualized by both synchrotron XPCT - with scalable FOV and resolution in the sense of multiscale imaging - and by laboratory μ-CT, and also present different quantification and visualization approaches. Further, we show that histological analysis can be performed after non-destructive XPCT enabling correlative imaging. The manuscript concludes with a Discussion section.

## Results

### 3D virtual histology and correlation with conventional 2D histology

Figure 3 shows virtual slices and 3D renderings of HFOs embedded in paraffin based on XPCT at the GINIX endstation using the SR-PB configuration, and compares a virtual slice through the tomographic reconstruction with an H&E-stained histological section. In order to correlate the virtual slice with the H&E-stained organoid section, shown in Fig. 3a, b, a manual alignment was performed in which specific structural features, i.e., different tissues, such as the shape of the endodermal islands in the OL or the endodermal cavities in the IC were identified within the reconstructed volume. The matching virtual slice was found in close proximity to the histological slice. To this end, registration was performed using Avizo Lite 9, in which planar virtual slices through the reconstructed volume can be created in any arbitrary direction. Small morphological differences between these two results, apart from inaccuracies in the manual image registration, can be attributed to slight distortions by re-embedding and slicing of the specimen, which was performed after tomographic imaging for histological analysis, as well as to the larger slice thickness (>5 μm) in the histological image.

The results show comparable morphological information, importantly, the three different layers IC, ML, and OL can be well distinguished in the virtual section. In contrast to conventional histology, non-invasive XPCT provides the structure of the HFO in its full three-dimensionality, exemplified in Fig. 3c by a volume rendering of orthogonal views of the reconstructed volume with isotropic resolution and a voxel size of 650 nm. The three layers contain characteristic tissue elements: The ML, which is separated from the OL on the outside and from the IC on the inside, is mainly composed of premature cardiomyocytes forming a dense network. Throughout the OL, dense cellular clusters, rosette-forming so-called endodermal islands, can be identified, harboring liver progenitors derived from the posterior foregut endoderm (PFE). Morphologically and functionally distinct from that, anterior foregut endoderm (AFE) tissue can be observed in the IC, which can be recognized by brick-like cells, i.e., columnar epithelium, forming larger cavities, especially in the outer regions of the IC, see for example Fig. 3d–f. In contrast to the PFE in the OL, the endodermal cavities represent anlagen for distinct organs including lungs, stomach and esophagus. In addition to endodermal cavities, the IC contains another type of cavities formed by single-cell endothelial layers, referred to as vessel-like structures, which share characteristics with native blood vessels. Both types of cavities can be well distinguished as shown in Fig. 3f. A distinct single-cell layer of endothelial cells surrounding the IC can be observed as well, Fig. 3g,

**Fig. 2 | Sample preparation and X-ray tomography setups. a** Successfully differentiated HFOs were first embedded in paraffin after dehydration and fixation of the HFOs. Biopsy punches with a diameter of 1 mm were then taken from paraffin embedded HFOs and placed into a Kapton tube. **b–d** Sketches of the tomographic setups and single measured projections, respectively. **b** Laboratory $\mu$-CT setup (EasyTom, RX Solutions, France) operated with a nanofocus transmission anode source and a CCD detector. **c** Sketch of the synchrotron radiation parallel beam geometry yielding an effective voxel size of 650 nm and (**d**) of the cone beam geometry of the GINIX setup at the P10 beamline at PETRA III (DESY, Hamburg). The cone beam geometry is realized by focusing the X-ray beam by a Kirkpatrick-Baez (KB) mirror, which is then coupled into a waveguide aligned in the focal plane of the KB mirror. This configuration enables high geometric magnification and effective voxel sizes below 200 nm.

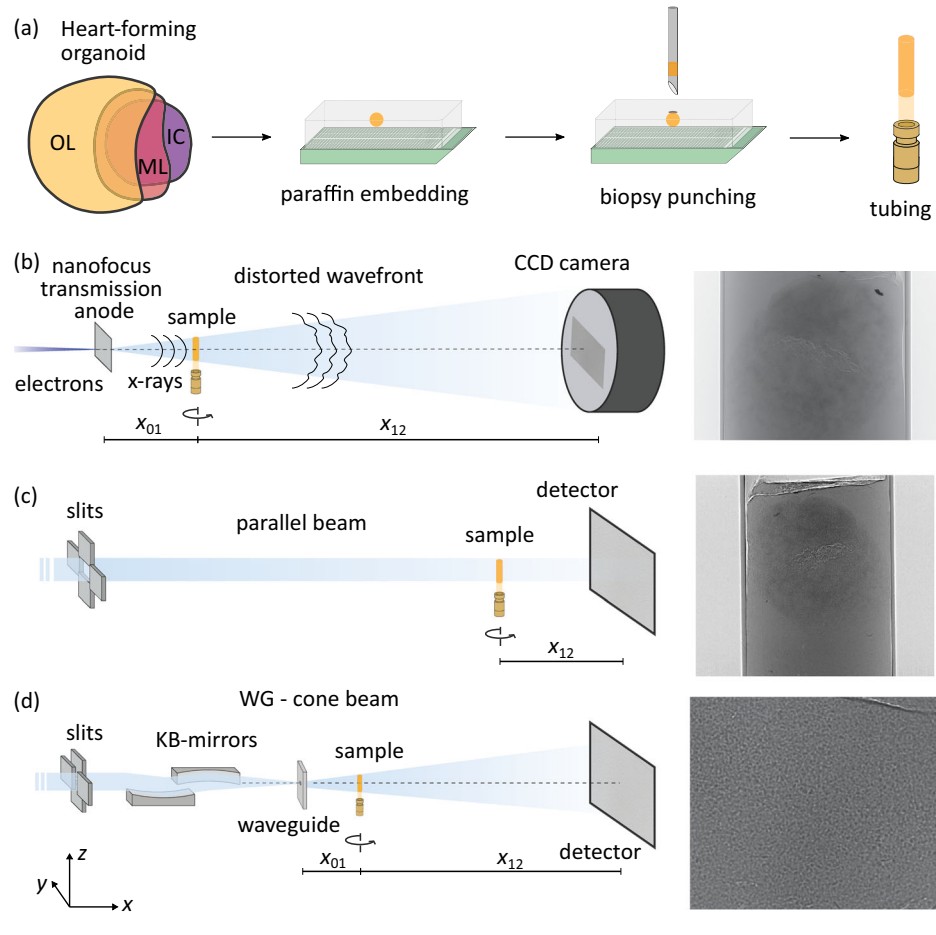

representing the formation of an endocardial-like layer, i.e., endocardium anlagen, between the ML and the IC.

Together, Fig. 3 showcases that XPCT in the parallel-beam geometry enables to obtain 3D structural information of important structures of interest already in the medium resolution range while the large FOV covers the tissue architecture of the whole organoid. The 3D structure of larger organoids within punch biopsies larger than 1 mm can be captured by scanning and stitching multiple tomograms as demonstrated in Supplementary Fig. 1. Note, that contrast formation in X-ray tomography relies on changes in electron density (approximately proportional to mass density), and morphology and structure is probed without any specific staining. The identification of different morphological features is mainly based on the thorough characterization of human HFOs previously published in ref. 21, but may also be based, at least in part, on morphology. To unequivocally resolve the differentiation or cell fate of single cells and different tissues, correlation with imaging techniques based on specific markers, such as immunohistochemistry, is required, since specific XPCT labels are not yet available.

## Laboratory-based $\mu$-CT

Next, we demonstrate the capability of a laboratory X-ray microscopy setup for 3D virtual histology of HFOs. Figure 4 shows tomographic results from two different FFPE HFO specimens obtained from a laboratory $\mu$-CT setup operated with a nanofocus transmission anode. The laboratory datasets are obtained with a similar FOV and effective voxel size (approximately 0.7 $\mu$m in the case of laboratory CT results) compared to those obtained with the parallel-beam setup at the synchrotron. The 3D representation and the representative virtual slice through the reconstructed volume correlated with an H&E stained histological slice in close proximity in Fig. 4a prove that the image quality of the laboratory dataset is sufficient to assess the 3D morphology of the organoid with relevant structural details.

The different layers of the HFO, the IC, the ML and the OL, can be distinguished as well as important differentiated cell types such as cardiomyocytes in the ML and different types of tissue such as liver anlagen (endodermal islands) in the OL can be observed. The image quality even allows to distinguish between the two different types of cavities - vessel-like structures and endodermal cavities - in the IC, see the comparison in the magnified views.

A direct comparison of the image quality between the laboratory dataset and the dataset obtained from the SR-PB configuration at the synchrotron is shown in Fig. 4b, c for the same HFO specimen. For this purpose, the magnified view in Fig. 4b based on laboratory CT results is compared to the corresponding virtual section in 4c for the same region based on the virtual slice presented in Fig. 3d resulting from the parallel beam configuration. In addition, a quantitative analysis of the image quality with respect to resolution and contrast-to-noise ratio (CNR) estimates are presented in Supplementary Figs. 5–7, both for SR-PB and laboratory $\mu$-CT reconstructions. For comparable effective voxel sizes similar values for the resolution were found, however, in particular contrast is superior when scanned with highly coherent synchrotron radiation allowing phase information to be retrieved from the measured projections. Nevertheless, the image quality of the laboratory dataset was found to be competitive and sufficient to identify important structural features of interest in an HFO, also at subcellular scale. Apart from the higher contrast, and from a practical point of view, a further important difference is given by the acquisition time. The lower photon flux (density) of laboratory X-ray sources compared to the synchrotron results in much longer acquisition times. In the present case scans took 8.7 h (Fig. 4a) and 21.8 h (Fig. 4b). This could potentially be reduced to a minimum of two hours (for given setup and samples), albeit at the cost of a lower signal-to-noise ratio, see Supplementary Fig. 2. For this reason, measurements of large sample series may be beneficially be carried out with synchrotron radiation.

**Fig. 3 | 3D virtual histology of HFOs by parallel-beam synchrotron radiation and correlative imaging. a** Light microscopy image of an H& E-stained histological slice of an HFO (HFO Sample 1), correlated to (**b**) a reconstructed tomographic slice in the xz-plane. Note, that H& E-staining and sectioning was performed after the tomographic recordings. **c** 3D representation of the reconstructed volume. **d** Reconstructed tomographic slice in the xy-plane and zoom into the inner core. **e** Volume rendering of endodermal cavities showing that they are partly connected throughout the IC. **f** Virtual slice (xz-plane) through a tomographic reconstruction obtained from a different HFO sample (HFO Sample 2) and correlation with an H& E-stained histological slice. The magnified views highlight vessel-like structures (purple and yellow circles) and an endodermal cavity (pink and green circles. **g** Virtual section through the IC at a different position along the y-direction shows a thin endocardial-like layer, black arrow in the magnified view.

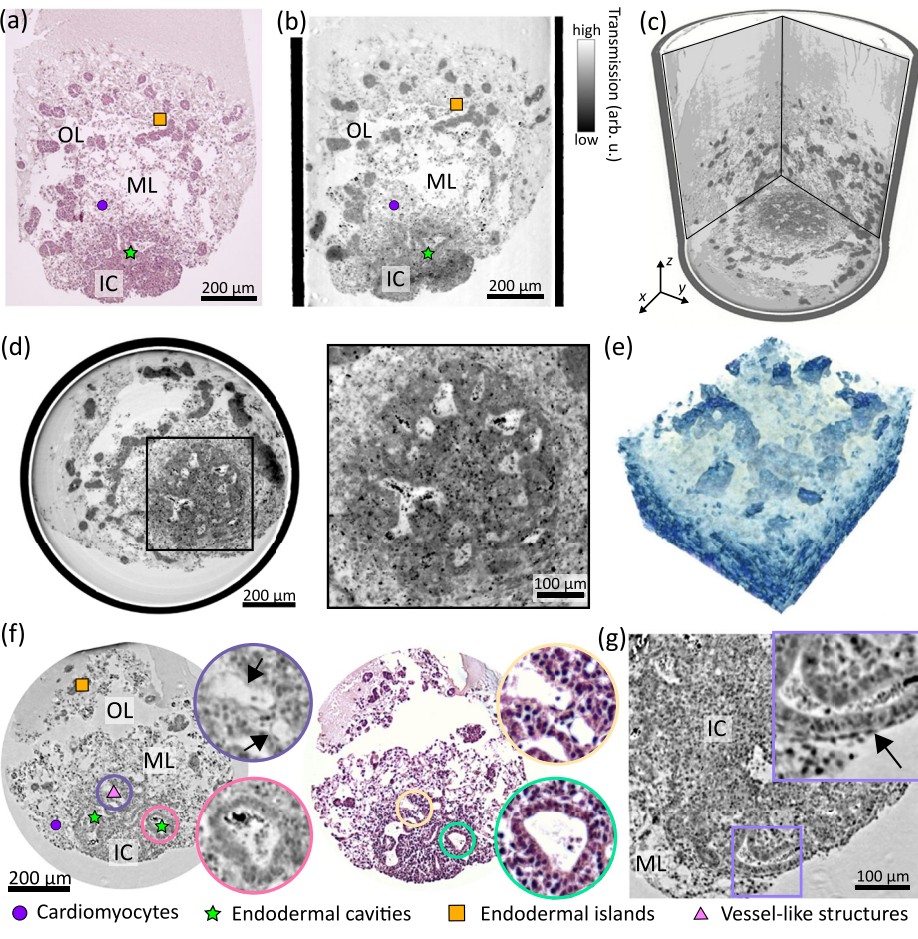

**Fig. 4 | 3D virtual histology of HFOs based on laboratory μ-CT. a** 3D representation (orthogonal slices) of the reconstructed volume from an HFO (HFO Sample 4) with an effective voxel size of ~ 0.7 μm, a virtual slice through the reconstructed volume and correlative histology (H& E) (registration and visualization with the viewer by Histomography GmbH). The different layers IC, ML, and OL are indicated in the virtual section. The zoom-ins (yellow and blue circles) highlight a vessel-like structure and an endodermal cavity. **b** Virtual slice through the reconstructed volume resulting from laboratory examination of the same HFO specimen as shown in Fig. 3a–e. The magnified view (green rectangles) shows a sub-region with the three layers of interest IC, ML, and OL. **c** Virtual section from the tomographic results of the SR-PB setup (cf. Fig. 3d) of the same region as shown in (**b**).

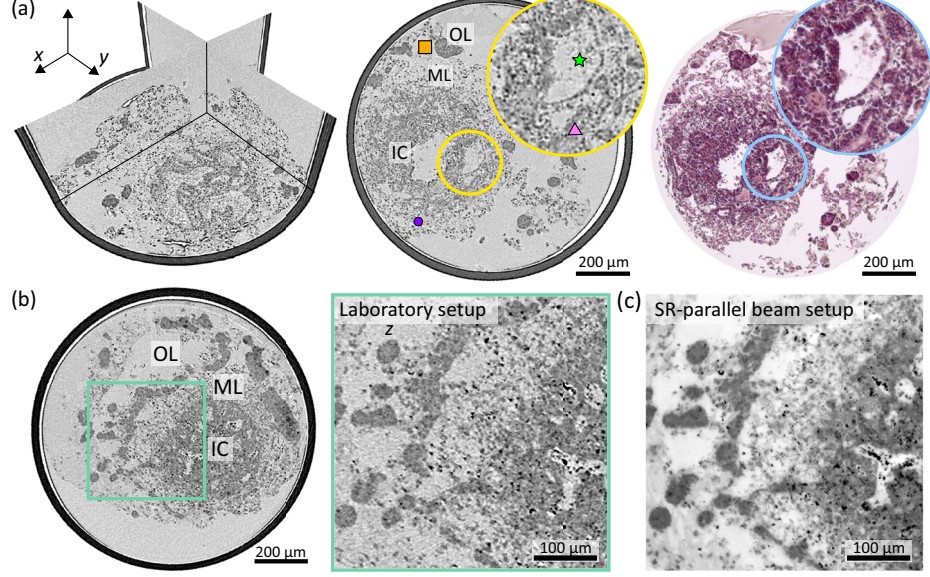

## High-resolution XPCT with X-ray waveguide optics

Figure 5 highlights results of the high-resolution synchrotron CT scan. As described above, the tomographic scan was recorded with a highly coherent cone-beam at 13.8 keV photon energy using an X-ray waveguide at the GINIX endstation at P10/PETRAIII. For orientation, the larger FOV probed by the laboratory CT setup (for same specimen as in Fig. 4a) is first shown in Fig. 5a. The subregion which is subsequently scanned at the high resolution setup (SR-CB) is marked by the black rectangle. Figure 5b then presents a view of orthogonal views and two representative virtual slices, which include parts of the IC and the ML with a FOV of $0.32 \times 0.32$ mm$^2$, at an effective voxel size of 127 nm. Note that slice 750 in (b) corresponds approximately to the region shown in the magnified view in (a) of the laboratory dataset.

The transition from the IC to the ML can be clearly identified, e.g., as indicated in Fig. 5b. Different structural features of interest are indicated and

**Fig. 5 | High-resolution XPCT of an HFO.**
**a** Virtual slice through the reconstructed volume of the laboratory data with a large FOV of HFO Sample 4. The magnified view marked by a black rectangle indicates the selected region for the high-resolution scan with X-ray waveguide optics shown in (**b**), corresponding to the same region as shown in slice number 750. The magnified view marked by a yellow rectangle shows a vessel-like structure corresponding to the vessel-like structure shown in (**b**, **c**) of the high-resolution scan. **b** 3D view into the reconstruction volume and representative virtual slices through the reconstruction volume obtained by high-resolution XPCT with X-ray waveguide optics showing part of the IC as well as a part of the ML. Different structural features such as cardiomyocytes, endodermal islands, endodermal cavities and vessel-like structures are indicated with the corresponding symbol. **c** Magnified views of a section of a vessel-like structure (left), an endodermal cavity (center), and the myocardial layer displaying a network of cardiomyocytes (right) are shown as maximum-intensity projections over 0.5 μm, corresponding to the marked squares in yellow, green and purple in (**a**), respectively. White arrows in (**a**, **c**) indicate an endothelial cell nucleus of the single-cell endothelial layer that forms the vessel-like structure.

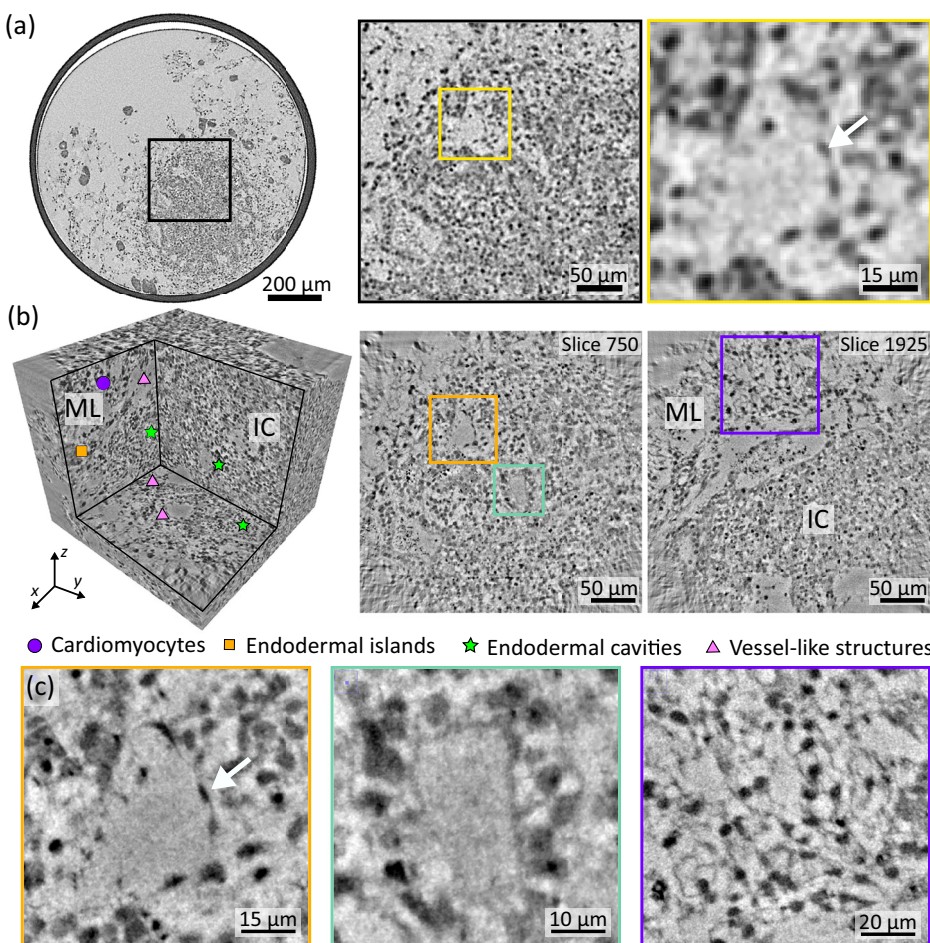

are additionally shown as magnified views in Fig. 5c corresponding to the marked squares in Fig. 5b. In the inner core, two types of cavities, the vessel-like structures and the endodermal cavities, are identified by the definitive morphology of the surrounding cells including their characteristic cell nuclei, as shown in the magnified views marked by orange and green squares, respectively. Compared to the vessel-like structure shown in (a) obtained by laboratory CT, finer (sub-)cellular details can be observed by high-resolution XPCT as demonstrated for the same region in (c). The vessel-like structures are lined by endothelial cells, which typically can be identified by their elongated morphology of both the cell and the cell nuclei. In contrast, the endodermal cavities are lined by brick-like epithelial cells. The ML surrounding the IC is mainly formed by a network of premature cardiomyocytes, shown in the magnified view (purple color). While the nuclei of the cardiomyocytes can be well resolved, substantial noise in the reconstruction and low contrast of the cardiomyocytes cell bodies limit the 3D structural information of their network-like arrangement. This observation is also supported by a quantitative assessment of image quality metrics. While the resolution is superior to that of the SR-PB and laboratory μ-CT setups, the CNR is found to be lower (see Supplementary Figs. 5–7). Although the contrast is sufficient to provide to some extent structural information about the cellular network, quantitative analysis of the network itself for example based on automated segmentation is still difficult at this stage. In future studies, contrast may be improved by recording the projections at multiple distances to account for zero crossings in the CTF phase reconstruction[23], resulting at the same time in a reduction in image noise by averaging the multiple projections at each angle.

## Segmentation and visualization of features of interest

The image quality of the high resolution tomographic reconstruction presented in Fig. 5, however, already enables the 3D segmentation and visualization of some features of interest. In a first example, we present the segmentation of the cardiomyocytes cell nuclei in the myocardial layer in Fig. 6.

The semi-automatic segmentation was performed by using the Blob Finder algorithm implemented in Arivis Vision4D (Zeiss AG, Germany), which is designed to find round, sphere-like objects, as e.g., recently applied to the segmentation of granule cell nuclei in the human cerebellum in an XPCT dataset[24]. A mask was manually drawn around the ML every few slices. The mask boundaries were then interpolated between slices in order to include only segments in the ML, also excluding reconstruction artifacts in the corners. From the 5914 segments found, different properties such as center of mass, volume, sphericity, mean gray values and standard deviation of gray values can be extracted. In (c) sphericity and volume are shown for all segments in a scatter plot. By visual inspection, the scatter plot shows a major cloud of data points for volumes between approximately 20 an 50 μm³ and around a sphericity of 0.6. Note that a sphericity of unity corresponds to an ideal sphere. Visualization of the segments properties by a scatter plot potentially supports the refinement of the segmentation: Small segments on the left, e.g., smaller than 10 μm³, can be regarded as false positives, while larger segments (with a lower sphericity) potentially contain 'double nuclei', i.e., two nuclei in close proximity regarded as one segment by the used segmentation input parameters. While the segmentation of cardiomyocyte nuclei presented here is considered illustrative, it is worth noting that 3D shape and density histograms - 3D distance histograms can also be computed - may be of histological interest in the study of different developmental or pathological states of the ML. When emphasizing this in future studies, a careful evaluation of the segmentation process and refinement of the histograms may be required.

Next, Fig. 7 illustrates the morphology of vessel-like structures including the splitting of capillaries indicating that angiogenesis occurs in HFOs in addition to vasculogenesis.

**Fig. 6 | Segmentation of the cardiomyocytes cell nuclei in the myocardial layer using the high-resolution tomographic dataset. a** A representative reconstructed slice is shown together with a mask (blue) enclosing the myocardial layer in which cell nuclei are then segmented. Endodermal islands as well as reconstruction artifacts in the corners are not considered. Note, that the grayscale is inverted compared to the previous figures. **b** Cell nuclei in a representative 2D-slice, with an overlay of segmented areas. **c** Scatter plot with each point representing a nucleus, at a position representing the segments properties volume and sphericity. **d** Three-dimensional rendering of the segmented nuclei.

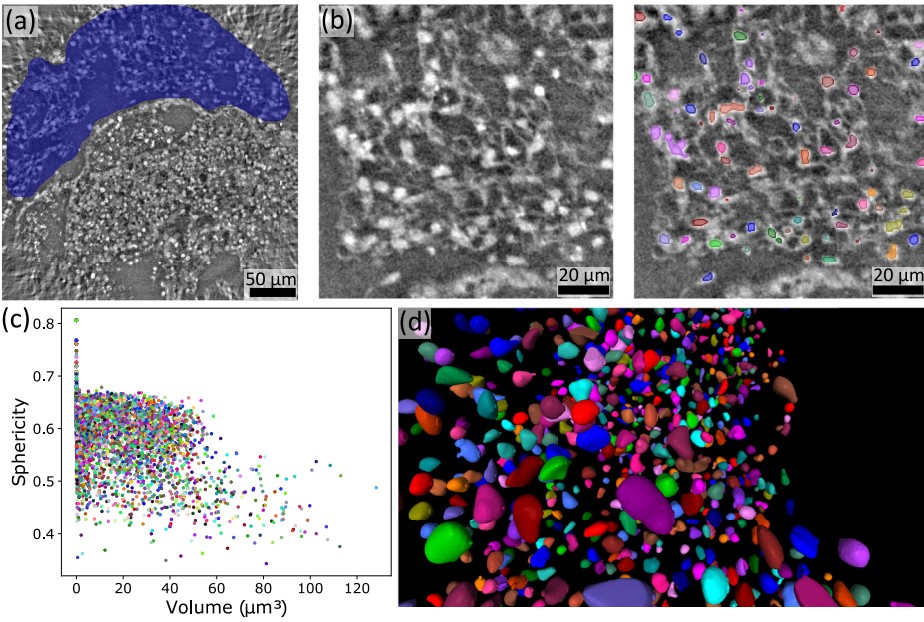

**Fig. 7 | Branching of vessel-like structures and segmentation. a** Magnified view of a vessel-like structure in a virtual section of a tomographic reconstruction (left) and in a H& E-stained section imaged by an optical microscope (right). **b** Volume-rendering of a manually segmented vessel-like structure. **c** Series of slices in z-direction to illustrate the separation of vessel-like structures. Yellow arrows mark the regions where separation of the capillaries takes place. Red arrows mark the single and the two separated vessel-like structures in slice 1178 and 1045, respectively. Scale bars: 15 μm in (**a**) and 10 μm in (**c**).

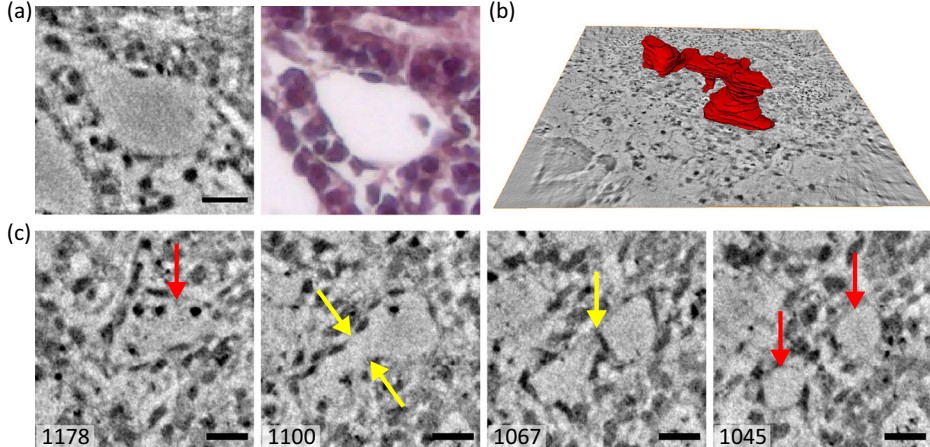

While cross-sectional vessel-like structures can be clearly identified in 2D histological sections in Fig. 7a, the separation of capillaries can only be observed by inspecting 3D volumes. Note, that the complete histological slice is shown in Supplementary Fig. 3. In Fig. 7b, the 3D rendering of a segmented vessel-like structure indicates multiple branching sites, yet at an early stage of development, resulting in an early developmental stage of a vascular network. The segmentation of the vessel-like structure was performed manually using the segmentation tools in Avizo Lite 9. The separation of a vessel-like capillary is illustrated in Fig. 7c: Following the series of slices, the vessel-like structure first elongates to an hourglass-shaped cross-section 9.9 μm above slice 1178 in slice 1100, and then begins to separate a further 4.2 μm above in slice 1067, until finally two separate capillaries are formed 2.8 μm further up in slice 1045. Inspection of the entire reconstructed volume reveals the formation of several independent vessel-like structures within the IC, which differ in size and morphology (beyond the examples presented here).

## Discussion

In summary, the present work demonstrates that propagation-based XPCT can be used for 3D tissue characterization of formalin-fixed and paraffin embedded (FFPE) human HFOs. The examples presented here show that the tissue architecture of the entire organoids as well as important structural features, i.e., of different types of tissue within the organoid, can be mapped in their full three dimensionality and at subcellular level. In contrast to conventional histology, XPCT provides digitized 3D volumes with isotropic resolution which can be virtually sliced in any arbitrary direction without physical sectioning. While first applications of XPCT as well as μ-CT on organoid systems have recently been reported[25–28], this work now provides the first comprehensive study on using multiscale XPCT for 3D screening and morphological analysis of FFPE organoids, combining scans recorded at different resolution and FOV. Importantly, we demonstrate that XPCT is compatible with the standard workflow for microscopic imaging of histological sections. The mm-sized punch biopsies of unstained FFPE organoids used for XPCT can be re-embedded into paraffin blocks for subsequent section-based histological investigation, enabling correlative imaging. While we demonstrated this for conventional histology based on H&E staining, immunohistochemical methods can be applied as well. The quality of histochemical and immunohistochemical staining is not affected by synchrotron-based XPCT, i.e., by typical dose depositions, as recently demonstrated on human FFPE tissue[29].

The multiscale approach enables large FOV and high throughput scans of entire punch biopsies using a parallel beam configuration and high resolution zoom scans at regions of interest based on cone beam magnification. Importantly, HFOs - but many other organoid systems as well - can

be produced in large quantities with a high reproducibility. For the present case, the mean success rate of about 88%[21] enables potential high-throughput applications such as drug screening in pharmacological research or disease modeling in fundamental biomedical research (exemplary, a first tomographic reconstruction of a *NKX2.5*-KO HFO is presented in Supplementary Fig. 4). A single large field of view ($1.6 \times 1.4$ mm$^2$) scan is acquired in less than two minutes, providing the 3D structure of the whole HFO in a 1 mm sized punch biopsy with an image quality sufficient to observe the different types of tissue such as endodermal islands and cavities, vessel-like structures as well as the cardiomyocyte network in the ML. Larger punch biopsies, here with a diameter of 3 mm, are covered by stitching several of overlapping scans, e.g., to capture the entire 3D structure of larger organoids. Punch biopsies of up to 8 mm in diameter can be feasibly scanned with the SR-PB setup, however image quality in terms of resolution and contrast generally decreases with increasing sample size and must be considered. Note that, even though not considered in this work, stitching of several scans potentially increases volume throughput when multiple organoids can be found within a single punch biopsy.

Sub-regions can be scanned at higher resolution by using the cone-beam setup based on nano-focusing for holographic imaging. In this work, this has been demonstrated by scanning a region that includes parts of the IC and the ML, providing the highly resolved 3D structure of the cardiomyocytes network, vessel-like structures and endodermal cavities. The magnification can generally be adjusted by the distance between the sample and the waveguide. Note, that there is always a trade-off between FOV and effective pixel size, i.e., a larger magnification results in a smaller effective pixel size and at the same time in a smaller FOV. In terms of high resolution, XPCT has been demonstrated at effective pixel sizes of well below 100 nm[30–32] and at resolutions down to 87 nm[31]. Notably, the resolution limits at this stage are primarily due to various factors within the imaging chain, such as the focusing X-ray optics, the mechanical stability of the sample stage and the image reconstruction methods, with recent advances in phase retrieval[33,34], for example, rather than the theoretical diffraction limit.

The image quality of the SR-CB data allows for the observation of fine structural details in 3D. The illustrative examples presented in this work showcase that quantitative information can be obtained through segmentation of structures of interest such as cell nuclei in the ML, which yields 3D shape and density information, as well as vessel-like structures. For the latter, segmented XPCT data enables to assess a detailed 3D analysis of the vascular network with the potential to elucidate early developmental stages of angio- and vasculogenesis, one of the earliest processes in embryonic organogenesis[35], in organoid model systems. Furthermore, 3D analysis of the vascular network opens the potential to study pathologies in such organoid models that are associated with their remodeling[36]. For example, pathological alterations of the vascularization have recently been analyzed quantitatively by XPCT in the context of virus-induced vascular remodeling (e.g., induced by COVID-19) in human lung and heart tissue[37,38].

Finally, we have shown that a laboratory $\mu$-CT equipped with a nano-focus X-ray source can be operated to match the SR-PB setup in FOVs and voxel size. The 3D reconstructions exhibit surprisingly rich structural information, and an image quality sufficient to identify the different layers IC, ML, and OL of the HFOs as well as important structures of different tissues in 3D. While the resolution was found to be similar (see Supplementary Figs. 5–6), in particular the CNR (see Supplementary Fig. 7) was significantly lower than for the SR-PB scans. Further, long acquisitions on the order of several hours are necessary. At the same time, the accessibility and commercial availability of laboratory setups and a possible operation in the direct vicinity of organoid growth and research are of significant advantage. Such instruments could be easily integrated in standard workflows of section-based microscopic imaging of FFPE organoids to obtain additional 3D information, in line with the discussion above with respect to the compatibility of XPCT with histological methods. The high image quality and contrast of modern laboratory $\mu$-CT setups in principle also allows for the segmentation of structures of interest, in particular high contrasting features such as cell nuclei, e.g., to compute distance histograms

in 3D, using for example (semi-)automatized gray value-based thresholding techniques[39] or machine learning approaches. Additionally, staining can be applied to biological tissues to enhance contrast of specific tissue regions[40], facilitating gray value - based segmentation in $\mu$-CT scans.

Regarding tissue preparation, we note that XPCT is not limited to paraffin-embedded samples. While the strength of FFPE samples clearly lies in their compatibility with established histological methods and very good image quality measures in XPCT, shrinkage of tissues in general, but also specifically of HFOs, has been observed during invasive fixation/dehydration and paraffin embedding (note, however, that the shape and tissue structure, including cavities, are very well preserved in HFOs when compared with results from complementary methods such as whole-mount immunofluorescence and cryosection staining[21]). In addition to paraffin, other embedding media have been used with the potential benefit of less invasive sample preparation, such as PBS or PBS/ethanol mixtures, but at the expense of image quality in terms of contrast and resolution[41,42]. Recently, also cryogenic fixation of lung tissue has been applied in XPCT as a promising method to study the tissue structure in its well preserved original state[12]. Considering as well potential drawbacks, future studies may carefully evaluate image quality and general applicability of different embedding media for organoid models.

All together, we have shown that both synchrotron-based XPCT and laboratory-based nano/$\mu$-CT provide valuable 3D structural information of HFOs, uniquely linking the subcellular- with the macro-scale with isotropic and scalable resolution. While we have focused on HFOs in this work, we further suggest that multiscale XPCT offers great potential for 3D imaging of other organoid systems as well, thus joining the toolset of volume imaging methods in organoid biology, importantly, also with high-throughput capabilities. Integrating 3D imaging by XPCT for novel organoid systems may be particularly useful to shed light on their morphogenesis and for further guidance. With ongoing progress in instrumentation, X-ray optics, automatization of sample handling and image reconstruction at dedicated synchrotron sources[43], both image quality and high-throughput capabilities can be expected to further improve in the foreseeable future. The same applies to laboratory sources: Higher sample-throughput and improved capabilities for quantitative image analysis due to an increase in image quality will be achieved by ongoing advances in the entire imaging chain, including in particular phase-retrieval capabilities. Future studies involving large organoid sample series, e.g., for biomedical applications, face the additional challenge of automated quantitative analysis, including image segmentation and statistical assessment of morphometric parameters.

## Methods
### Formation and culture of HFOs
Experiments using hESC lines are performed under allowance '108: Genehmigung nach dem Stammzellgesetz' granted by the Robert Koch Institute (Berlin, Germany). The following cell lines were used: HES3 NKX2.5-eGFP[44] and HES3 NKX2.5-eGFP/eGFP (NKX2.5-KO)[45]. HFOs were formed within a 14-day protocol as described in detail in refs. 21,22. Briefly, within an initial 4-day preculture period, hPSC aggregates are individually generated in a 96-well format and subsequently embedded in Matrigel droplets. Thereafter, directed differentiation is induced by applying the WNT pathway modulators CHIR99021 and IWP2 in a time- and concentration-dependent manner.

### Sample preparation for X-ray tomographic recordings and histology
Organoids were fixed with 4% PFA at 4 °C overnight, washed three times with phosphate-buffered saline (PBS) and dehydrated and embedded in paraffin using the following protocol at the HistoCore PEARL (Leica): Dehydration was performed by ethanol series (70%, 80%, 95%) at 45 °C for 40 min each, followed by three 1 h incubations in 100% Ethanol and three 1 h incubations in RotiClear at 45 °C. Afterwards, the samples were incubated three times in paraffin at 65 °C (for 1.5 h, 2 h and 2.5 h). The

paraffinized organoids were then transferred to metal embedding molds, which were filled with hot, liquid paraffin, and finally cooled down to form a paraffin block.

For X-ray tomographic recordings, HFOs were punched out by using a biopsy punch either with a diameter of one or three mm, resulting in a cylindrical shape of the specimen which is beneficial for the final image quality. The biopsy punches were then mounted on a brass pin, either free-standing in the case of 3 mm punches or placed into a polyimide (Kapton) tube for stabilization of 1 mm punches, see Fig. 2. For subsequent histological analysis, the HFOs within the biopsy punches were re-embedded into paraffin blocks. From these blocks, sections of 3 μm in thickness were generated using the Leica RM2245 microtome (Leica). Tissue sections were stained with H&E using standard protocols.

### Propagation-based X-ray phase-contrast tomography

Tomographic datasets were acquired by means of propagation-based XPCT at the Göttingen Instrument for Nano-Imaging with X-Rays (GINIX) holotomography endstation of the P10 undulator beamline (Petra III, DESY, Hamburg, Germany)[46]. At this instrument, two different optical configurations, schematically shown in Fig. 2, were used for multiscale imaging, which are implemented in close proximity[47]: the SR-PB setup and the SR-CB setup, where the sample is illuminated by a quasi-point source created by focusing mirrors and an X-ray waveguide. Since the first publication in ref. [47], the GINIX imaging workflow, detectors, data acquisition and instrument control, including continuous rotation of the tomographic axis and tools for easy configuration of complex high-dimensional measurements, e.g., for stitching 3D tomograms, have been continuously optimized. Recently introduced germanium waveguides were also used, providing enhanced holographic illumination.

X-ray tomographic recordings with relatively large field-of-views of approximately $1.5 \times 1.5$ mm$^2$ were acquired by the SR-PB setup. The beam size of the unfocused and quasi-parallel beam was adjusted by upstream slits to approximately $2 \times 2$ mm$^2$. Well defined photon energies of 8, 11, and 13.8 keV were selected by a Si(111) channel-cut monochromator. For a single tomogram, 3000 projections were recorded over a continuous rotation of 360°. A high resolution microscope detection system (Optique Peter, France) was used based on a 50 μm thick LuAG:CE scintillator and a 10x magnifying microscope objective coupled to a sCMOS sensor (pco.edge 5.5, PCO, Germany) with a 100 Hz maximum frame rate, a rolling shutter and a fast scan mode. Higher magnification lenses, such as 20x, are available and give smaller effective pixel sizes. In practice, however, increased magnification does not necessarily translate into a significant gain in resolution, which is limited by a number of other factors. Therefore, the 10x lens is a suitable compromise, providing a larger FOV without significant loss of effective resolution. The high photon flux density enabled a short acquisition time of 35 ms per frame, resulting in a total acquisition time of approximately 105 s for a single tomographic scan. The sample-to-detector distance was set to $x_{12} = 33$ mm; the pixel size was px = 0.65 μm. Details of the SR-PB setup are described in ref. [47]. The acquisition parameters for each sample are summarized in Table 1.

High-resolution X-ray tomographic scans were recorded based on geometric magnification by cone-beam illumination (SR-CB setup). After passing the upstream slit system, the X-ray beam was focused by the Kirkpatrick-Baez (KB) mirrors to approximately 300 nm, and then coupled into an X-ray waveguide resulting in a further reduction of the beam size with a sub-50 nm spot size, high spatial coherence and a filtered wavefront. For the photon energy of 13.8 keV, a germanium waveguide was used with 200 mm optical depth and a channel diameter of 100 nm. For a single tomogram, 1500 projections were recorded with a fiber-coupled sCMOS camera (Zyla 5.5 HF, Andor) with $2560 \times 2160$ pixels of 6.5 μm pixel size. With a source-to-sample distance of $x_{12} = 0.1$ m, sample-to-detector distance $x_{12} = 5.1$ m, the projection images were magnified at $M = 51.1$ with effective pixel size px$_{\text{eff}} = 0.1272$ μm. Together, these parameters resulted in image acquisition in the so-called deeply holographic regime, with a Fresnel number of $F = \text{px}_{\text{eff}}^2/(x_{12}\lambda) = 0.0018$.

**Table 1 | Experimental parameters**

| | SR-PB setup | | | SR-CB setup |
|---|---|---|---|---|
| | HFO Sample 1 | HFO Sample 2 and Sample 3 | HFO NKX2.5-KO | HFO Sample 4 |
| No. projections | 3000 | 3000 | 3000 | 1500 |
| FOV ($h \times v$) (mm$^2$) | $1.6 \times 1.4$ | $1.6 \times 1.4$ | $1.6 \times 1.4$ | $0.32 \times 0.27$ |
| $x_{01}$ (m) | 88 | 88 | 88 | 0.1002 |
| $x_{12}$ (m) | 0.033 | 0.022 | 0.022 | 5.01 |
| $px_{\text{eff}}$ (nm) | 650 | 650 | 650 | 127 |
| Exposure time (s) | 0.035 | 0.035 | 0.035 | 1 |
| Total exposure time (s) | 75 | 75 | 75 | 1500 |
| Energy (keV) | 8.08 | 11 | 13.8 | 13.8 |
| Fresnel Number | 0.0834 | 0.1622 | 0.2138 | $1.8 \times 10^{-3}$ |
| $\delta/\beta$ | 65 | 65 | 65 | 65 |
| $\alpha_{\text{low−freq}}$ | 0 | 0 | 0 | 0.005 |
| $\alpha_{\text{high−freq}}$ | 0.2 | 0.2 | 0.2 | 0.1 |

Acquisition and reconstruction parameters for the SR-PB and SR-CB setups and for the different HFO samples. $x_{01}$ denotes the source-to-sample distance, $x_{12}$ the sample-to-detector distance, $px_{\text{eff}}$ the effective pixel size, $F$ the Fresnel number, $h$ and $v$ the horizontal and vertical length, respectively, $\delta/\beta$ the ratio between refraction and attenuation in the sample, and $\alpha_{\text{low−freq}}/\alpha_{\text{high−freq}}$ the lower and the upper cut-off frequency, respectively, for the CTF/NLT reconstruction algorithm.

### Phase retrieval and tomographic reconstruction

Prior to phase retrieval, projections were corrected for flat field and dark images. Phase retrieval was then performed either by the single-step contrast-transfer-function (CTF) approach[23] (SR-PB data), based on the linearisation of optical constants, or by the non-linear Tikhonov (NLT) algorithm[48,49] (SR-CB data), a non-linear generalization of the CTF reconstruction method that accounts for optically non-weak objects, both implemented in the Holotomo-Toolbox[49]. In both cases an object of homogeneous composition was assumed, i.e., a coupled phase shift and attenuation, here set to the ratio $\delta/\beta = 65$ for each phase reconstruction, see also Table 1 for a summary of the reconstruction parameters. After phase retrieval, tomographic reconstruction was performed by filtered back-projection as implemented in the ASTRA-Toolbox[50] and incorporated in the Holotomo-Toolbox.

### Laboratory X-ray μ-CT setup

The samples were scanned using a laboratory μ-CT (EasyTom, RX Solutions, France), schematically shown in Fig. 2b, equipped with a nanofocus transmission anode x-ray source (Hamamatsu L10711-02, W target, LaB6 cathode), a CCD camera (Gadox scintillator with free fiber optics coupling, $4008 \times 1704$ pixels, 9 μm pixel size), and a custom-made sample holder. Tomographic scans were collected at high geometric magnification by moving the sample close to the source spot. For tomographic reconstruction, 1568 projection images were recorded over 360° with a tube voltage of 60 kV. To improve the signal-to-noise ratio (SNR), up to 12 projection images were accumulated for each projection angle, i. e. at the same position of the sample, with an exposure time of 2 or 5 s for a single frame. Reference images at selected angles for subsequent tracking and correction of spot size movement over the duration of the scan, as well as flat and dark field images for flat field correction were recorded prior to the tomographic scan. Ring filter correction, a Fourier-filter reducing edge enhancement (simplified phase retrieval supplied with the instrument software), and tomographic cone-beam reconstruction were performed with an instrument-specific software.

### Visualization and segmentation

The Fiji open-source software[51] and MATLAB (R2022b) were used for further analysis and visualization, and NVIDIA IndeX (NVIDIA, Santa

Clara, US)[52] and Avizo (Thermo Fisher Scientific, Waltham, US) for volume renderings. For segmentation of cell nuclei in the myocardial layer, the software Arivis Vision4D (Zeiss AG, Germany) was used in a semiautomatic workflow using the Blobfinder algorithm as described in detail in ref. 24. The adjustable input parameters required for the segmentation using the Blobfinder algorithm were "Diameter": 3.6 μm, "Probability Threshold": 22% and "Split Sensitivity": 40%, and were set manually based on visual inspection. Segmentation of vessel-like structures was performed manually using Avizo (Thermo Fisher Scientific, Waltham, US). In general, the ability to trace the cardiomyocytes cell nuclei and vessel-like structures by (semi-) manual segmentation is based on visual inspection of the image data and segmentation results, and relies on the image quality and the observers experience.

## Statistics and reproducibility

3D visualization using different X-ray phase-contrast tomography configurations of HFOs, including representative features of interest such as the characteristic layered pattern (IC, ML, and OL), different tissue components and vascularization, required only a limited number of HFOs. To this end, four HFOs and one NKX2.5-knockout HFO were imaged, of which three HFOs were used for further H&E-histological analysis to enable correlative imaging. All HFOs displaying the typical NKX2.5-eGFP layered pattern were considered to be successfully formed. Only these successfully formed HFOs were used in this study (see ref. 21, for details). No statistical analyses of the data were required for this study.

## Reporting summary

Further information on research design is available in the Nature Portfolio Reporting Summary linked to this article.

## Data availability

Raw data generated at DESY will be released and made public two years after the beamtime. All treated datasets are available from the corresponding authors on reasonable request. Exemplary datasets that support the findings of this study are openly available in GRO.data (https://doi.org/10.25625/UWPLSQ). The source data underlying the graph in Fig. 6c is available in Supplementary Data 1.

## Code availability

Phase retrieval and tomographic reconstruction were performed by using the Holotomo-Toolbox[49] and the ASTRA-Toolbox[50] as incorporated in the Holotomo-Toolbox. Matlab scripts for performing the reconstruction are available from the corresponding authors on reasonable request.

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

## Acknowledgements
We thank Lilli Geworski for her support. We acknowledge DESY (Hamburg, Germany), a member of the Helmholtz Association HGF, for the provision of experimental facilities. Beamtime was allocated for proposal I-20230026. We would also like to thank Michael Sprung, Wojciech Roseker, Fabian Westermeier and Markus Osterhoff for their continuous support at the GINIX instrument at beamline P10 (PETRA III, DESY). We thank Jette Alfken and Jannis Schaeper for support during beamtime, Jana Teske and Lara Treumann for support in histological preparation and imaging, and Jakob Frost for support in image segmentation. Financial support from the German Ministry for Research, Technology, and Space for grants HoToP4 (05K25MG2) and Holo-Tomograhy (05K22MG1) within the ErUM-Pro funding line, the German Research Foundation (DFG) in the framework of the Germany's Excellence Strategy EXC 2067/1-390729940, the Ministry of Science and Culture of the State of Lower Saxony (MWK; program "Zukunft.Niedersachsen"; grant ZN4092), the European Union through the European Regional Development Fund (EFRE) and the program area of the State of Lower Saxony "Stärker entwickelte Region" (SER); funding period 2021-2027 (ZW 3 87035144, project QUADRANT) is gratefully acknowledged.

## Author contributions
All authors designed research. K.K., J.R., and T.S. conceived and conducted the synchrotron experiments. K.K. and J.R. performed the laboratory $\mu$-CT experiments. L.D. and R.Z. selected the samples and L.D. prepared the samples. K.K. performed data processing, segmentation and visualization with support from J.R. L.D. performed histological preparation and imaging. K.K. and T.S. wrote the manuscript. All authors reviewed the manuscript.

## Funding

## Competing interests
The authors declare no competing interests.
