## [Transparent Peer Review file · Communications Biology]

3D histology of human heart-forming organoids by X-ray phase-contrast tomography

Corresponding Author: Dr Karlo Komorowski

Version 0:

Reviewer comments:

Reviewer #1

(Remarks to the Author)

The authors present the use of an X-Ray based PC-CT approach to image heart forming organoids, to identify the structures typically present in these samples, without the need for classical section-based histology. The technology presented is undergoing significant developments and has been applied to a number of different biological samples. In my opinion the novelty of this article is the application of this technology to organoids with such a high structural heterogeneity. Overall, I believe this article is presenting interesting results that will be useful for the field and I would like to recommend its acceptance with minor revision. I have therefore a few comments that I would like the authors to address:

In the method section, it should be highlighted if and how the samples have been dehydrated. Dehydration is problematic for other types of organoids (i.e. cystic lung / renal organoids), where it's affecting organoids shape, size, and causes ruptures. In my opinion this should be mentioned in the discussion section.

In Figure 3, I believe a histological section presenting the Vessel-Like structures, would be beneficial for the comparison of how these structures look in the X-Ray-based image

In figure 4, endodermal cavities and vessel like structures appear very similar. Can you please indicate how this distinction has been implemented?

In figure 5, please provide a better identification of the vessel like structures, possibly using arrowheads. I am still struggling to see the difference between these and the endodermal islands.

Figure 6: Can the authors provide supporting evidence that the segmentation presented in can be trusted? Since this is based on interpolated manual sectioning I think it would be beneficial to provide more insights on the process for the sake of reproducibility.

It would be interesting to understand the authors perspective on the value of the lab-based scans, since segmentation itself seems already to be difficult on highest quality scans acquired at the synchrotron. I agree that this topic is already discussed but I would possible like to see it expanded in the final version of the manuscript.

Reviewer #2

(Remarks to the Author)

This manuscript presents an innovative approach for performing non-invasive 3D histological analysis of heart-forming organoids using X-ray phase-contrast tomography (XPCT). The authors demonstrate the workflow and results of applying XPCT to heart-forming organoids, comparing the advantages and disadvantages of three different XPCT methods. Given the growing interest in the field of 3D histology and the potential for using this technology for complete tissue sample analysis, the study is of considerable relevance. However, there are several important issues that should be addressed in a substantial revision before it can be published.

Major Comments:

1. The technique presented in this manuscript appears to be similar to that described in a previous publication by the authors (Reference 26). Could the authors clearly highlight the specific innovations in the imaging technology presented here?
2. The manuscript does not provide sufficient details about the tomography reconstruction algorithm used. As tomography reconstruction is a critical aspect of XPCT, a clear explanation of the algorithm, including any optimizations or novel approaches, is essential for understanding the overall technical merit of this study.
3. The manuscript lacks a clear explanation regarding the differences between SR-PB and SR-CB modalities. Specifically, the authors should clarify why SR-PB mode is chosen for large field-of-view imaging and SR-CB for high-resolution imaging. It would be useful to provide a detailed rationale for this choice, especially considering the potential for using a higher magnification objective in SR-PB mode. Could the authors demonstrate whether using a higher magnification objective in SR-PB mode could achieve similar high-resolution imaging as SR-CB mode? Addressing this question would help readers understand the technical rationale and limitations of each modality more thoroughly.
4. The manuscript claims the ability to achieve isotropic resolution, yet I did not find any experiments or results related to resolution measurement or characterization. The paper should include experimental validation of isotropic resolution and provide specific quantitative metrics, such as resolution measurements, in addition to the voxel size.
5. Do successive rotations of the sample produce motion artifacts during rapid imaging?
6. In the supplementary materials, the authors present imaging results of a 3 mm-sized punch biopsy using a stitching algorithm. I am curious whether this method can be applied to larger sample sizes. Specifically, what is the maximum sample size that this approach can support without requiring tissue clearing treatments? Additionally, could the authors discuss whether the sample size might affect the imaging accuracy or resolution? Understanding whether there are any limitations in terms of sample size and how these factors impact the overall imaging performance would provide valuable insight into the scalability and practical applicability of this technique.
7. In the imaging results section, there is insufficient comparison with conventional histology based on H&E staining. For example, in Figures 3–5, the authors distinguish different tissue regions and cellular structures using XPCT. However, it would strengthen the manuscript to further compare these results with corresponding H&E staining images to validate and demonstrate the accuracy of the XPCT method. This comparison would provide a more robust assessment of the effectiveness of XPCT for tissue characterization, helping to establish its reliability as a non-invasive alternative to conventional histology.
8. The manuscript lacks detailed explanation and comparison regarding the identification and segmentation of features of interest. For instance, in Figure 6c, the purpose and biological significance of the scatter plot are unclear. The authors seem to focus solely on analyzing the accuracy of the segmentation algorithm without elucidating what biological information is being conveyed. Additionally, it is unclear whether the accuracy issues are due to image quality or the segmentation algorithm itself. A comparison of the method's accuracy with traditional HE staining would be beneficial. Furthermore, the authors should emphasize and compare the functionalities and applications that their method enables, which are challenging or unattainable with conventional H&E methods. Providing this information would significantly enhance the understanding of the advantages and potential applications of the XPCT technique in biological research.

Minor Comments:

1. Lack of scale bars in figures. Many figures lack scale bars, especially the 3D illustrations.
2. The rendering of Figure 7.b is not aesthetic, and it seems that only the segmentation results of blood vessels are displayed in 3D, while the original data only shows a single 2D plane. Why is it presented in this manner?

Reviewer #3

(Remarks to the Author)

The manuscript focuses on using X-ray phase-contrast tomography (XPCT) for three-dimensional (3D) histology of human heart-forming organoids (HFOs). These organoids are derived from human pluripotent stem cells and model early heart and vasculature development. The authors detail the application of XPCT for visualizing HFOs in their full 3D complexity without destructive slicing, offering high-resolution insights into organoid structure. They also compare synchrotron-based XPCT and laboratory micro-CT (μ CT) for imaging capabilities, segmentation, and visualization. The work demonstrates the utility of XPCT in complementing traditional histology and highlights its potential for high-throughput applications in developmental biology and regenerative medicine.

The manuscript is well-structured, addressing a critical limitation in organoid imaging by presenting XPCT as a non-invasive method for obtaining high-resolution 3D structural data. It integrates innovative imaging techniques with biological applications, advancing both technical and biological fields. The study is significant for organoid-based research, offering a scalable and isotropic approach to tissue visualization. However, the manuscript could benefit from more quantitative validation of imaging data, clearer delineation of the advantages of XPCT over μ CT in practical applications, and a broader discussion of limitations and future improvements.

I recommend this paper for publication if it can address the following questions outlined below:

Questions:

1. What specific quantitative metrics, such as contrast-to-noise ratios, spatial resolution, or signal uniformity, can be presented to rigorously compare the performance of XPCT and μ CT in imaging organoid structures?
2. How does the proposed XPCT methodology address challenges in scalability, particularly for high-throughput imaging of large datasets or multiple organoid samples, in terms of acquisition speed and data processing?
3. What are the fundamental limitations of XPCT in its current implementation, particularly concerning phase retrieval?

accuracy, resolution trade-offs, or compatibility with live-cell imaging?

Version 1:

Reviewer comments:

Reviewer #1

(Remarks to the Author)

The authors have responded to my comments and clarified all my doubts on the imaging parameters / anatomical structures identification. I am satisfied by the current version of the manuscript.

Reviewer #2

(Remarks to the Author)

I appreciate the authors' responses to my previous comments. The revised manuscript shows clear improvements in both clarity and technical depth.

The additional data and clarifications concerning the imaging methodology and its integration with biological interpretation are helpful and address earlier concerns. In particular, the inclusion of comparative analysis with H&E staining strengthens the validation of the XPCT approach. The expanded description of the computational pipeline and clarification of resolution-related parameters further improve the transparency of the work.

Overall, I find the revisions satisfactory and believe the manuscript is now suitable for publication.

Reviewer #3

(Remarks to the Author)

Thank you for the revision.

I have carefully reviewed the authors' responses and the revised manuscript.

The authors have adequately addressed my previous concerns, and I find the manuscript has improved significantly.

I have no further comments.

I recommend the manuscript for publication in its current form.

Reply and Summary of Changes

(Manuscript #: COMMSBIO-24-4012A)

We would like to thank the reviewers for their constructive feedback and helpful comments. Based on these, we have thoroughly revised the manuscript, which has undoubtedly been improved. Our point-by-point response is provided below (in blue). Please note that the updated or new figures can be found at the end of this response letter, for both the revised main article and the supplementary information document as indicated.

Reviewer #1 (Remarks to the Author):

The authors present the use of an X-Ray based PC-CT approach to image heart forming organoids, to identify the structures typically present in these samples, without the need for classical section-based histology. The technology presented is undergoing significant developments and has been applied to a number of different biological samples. In my opinion the novelty of this article is the application of this technology to organoids with such a high structural heterogeneity. Overall, I believe this article is presenting interesting results that will be useful for the field and I would like to recommend its acceptance with minor revision. I have therefore a few comments that I would like the authors to address:

Reply: We thank the reviewer for his/her positive assessment.

In the method section, it should be highlighted if and how the samples have been dehydrated. Dehydration is problematic for other types of organoids (i.e. cystic lung / renal organoids), where it's affecting organoids shape, size, and causes ruptures. In my opinion this should be mentioned in the discussion section.

Reply: The organoid samples have been dehydrated through an ascending series of ethanol for paraffin embedding. We have now added the information in the *Materials and Methods* section accordingly (line 122). Applying our fixation/dehydration protocol, we do observe shrinkage of the organoids; however, the shape and tissue structure including cavities are preserved very well, which we assessed via comparison to results from complementary methods such as whole-mount immunofluorescence and cryosection staining (Ref. 21: Drakhlis et al., Nat. Biotechnol., 2021). We now discuss this in the *Summary, Conclusions and Outlook* section (line 476).

In Figure 3, I believe a histological section presenting the Vessel-Like structures, would be beneficial for the comparison of how these structures look in the X-Ray-based image

Reply: We have now included additional correlative histology based on H&E histology highlighting both vessel-like structures and endodermal cavities. To this end, we replaced sub-figure (f) by a comparison between a virtual section through the tomographic reconstruction with an H&E stained histological section.

In figure 4, endodermal cavities and vessel like structures appear very similar. Can you please indicate how this distinction has been implemented?

Reply: The distinction is based on the morphology of the characteristic cells (and nuclei) that surround the cavities. While endodermal cavities are surrounded by “columnar epithelium”, vessel-like structures are made up of a single-cell endothelial layer, which we now state explicitly in the *Results and Discussion* section when introducing Fig. 3 (lines 254-260). To some extent, the size of the cavities can also provide a further clue, as endodermal cavities are often larger than vessel-like structures. We note, however, that the distinction can sometimes be made more difficult by examining only a single 2D slice, and that examination of the entire 3D volume, i.e. several successive slices, often gives a clearer picture. Importantly, we have also modified Fig. 4(a) of the revised manuscript to include an H&E histological section correlated with a tomographic section, highlighting a vessel-like structure and an endodermal cavity.

In figure 5, please provide a better identification of the vessel like structures, possibly using arrowheads. I am still struggling to see the difference between these and the endodermal islands.

Reply: Accordingly, we have added arrowheads pointing to an endothelial cell nucleus in the single cell endothelial layer. Together with the changes made in Figures 3 and 4, we hope that the distinction between vessel-like structures and endodermal cavities is now clearer.

Figure 6: Can the authors provide supporting evidence that the segmentation presented in can be trusted? Since this is based on interpolated manual sectioning I think it would be beneficial to provide more insights on the process for the sake of reproducibility.

Reply: The semi-automatic segmentation of the cell nuclei using the Blobfinder algorithm involves a number of steps that are based on the visual (and manual) inspection of the observer, so that the experience of the observer indeed plays a role. First, a mask is manually drawn to include only the nuclei of the ML, while excluding other parts such as the IC or other morphological structures such as endodermal islands. The interpolation of the mask between the drawn sections is then visually inspected. The Blobfinder algorithm itself requires adjustable input parameters, notably "diameter", "probability threshold" and "split sensitivity", which affect the size and number of objects found, and are set manually based on visual inspection. We have now revised the *Materials and Methods* section under “Visualization and segmentation” to include the above-mentioned information for greater clarity (lines 219-226).

It would be interesting to understand the authors perspective on the value of the lab-based scans, since segmentation itself seems already to be difficult on highest quality scans acquired at the synchrotron. I agree that this topic is already discussed but I would possible like to see it expanded in the final version of the manuscript.

Reply: We have added two more sentences together with Ref. 48 and 49 in the *Summary, Conclusions and Outlook* section (line 469, 500), additionally highlighting that segmentation of structures of interest in laboratory-based reconstructions is generally possible and discussing future directions (with the emphasis on phase-retrieval, which played only a minor role in the laboratory-based scans in this work). We expect that phase retrieval more specifically matching the conditions of μ -CT can bring future improvements in image quality.

Reviewer #2 (Remarks to the Author):

This manuscript presents an innovative approach for performing non-invasive 3D histological analysis of heart-forming organoids using X-ray phase-contrast tomography (XPCT). The authors demonstrate the workflow and results of applying XPCT to heart-forming organoids, comparing the advantages and disadvantages of three different XPCT methods. Given the growing interest in the field of 3D histology and the potential for using this technology for complete tissue sample analysis, the study is of considerable relevance. However, there are several important issues that should be addressed in a substantial revision before it can be published.

Reply: We thank the reviewer for the positive assessment and that she or he believes that the study “is of considerable relevance”.

Major Comments:

1. The technique presented in this manuscript appears to be similar to that described in a previous publication by the authors (Reference 26). Could the authors clearly highlight the specific innovations in the imaging technology presented here?

Reply: We agree that we follow the approach of Ref. 26 in the combination of parallel beam and cone beam phase contrast tomography. Since first published in 2020, we have continuously improved the image chain, detectors, data acquisition, and instrumental control, including the continuous rotation of the tomographic axis. We now have expanded the available set of waveguides which offer improved holographic illumination, for example the Germanium waveguide which was previously not yet available. This and the other technical improvements of the setup are now explicitly mentioned. Note however, that in this manuscript the specific innovations in imaging technology per se, i.e. in the hardware and software of the imaging setups or in reconstruction algorithms, were not the main focus. In this study, we provide the first comprehensive characterization of an organoid model (here using the 'multi-tissue' heart-forming organoid) by multiscale XPCT, combining recent advances in X-ray optics, instrumentation and phase retrieval, see e.g. (Ref. 26, 28, 29). Furthermore, we demonstrate that the translation from synchrotron to laboratory-based X-ray CT is feasible, making 3D histology of organoids accessible to a wider community. To date, the applicability of (multiscale) XPCT to organoid models has been largely unexplored.

2. The manuscript does not provide sufficient details about the tomography reconstruction algorithm used. As tomography reconstruction is a critical aspect of XPCT, a clear explanation of the algorithm, including any optimizations or novel approaches, is essential for understanding the overall technical merit of this study.

Reply: We have revised the manuscript in order to disclose the full details of the image reconstruction pipeline (lines 189-195 and Tab. 1), including in particular phase retrieval of the projections. The most advanced algorithmic component is certainly the non-linear phase retrieval based on iterative Tikhonov regularization described in detail in Huhn et al (Ref. 28). The tomography itself is a state-of-the-art implementation offered by the open domain astra toolbox (Ref. 30). Here, we chose the filtered-back-projection (FBP), since the cone angle was small enough. We have tested the Feldkamp-Davis-Kress (FDK) algorithm as an alternative, but essentially obtain identical results. Several smaller tools are available at the instrument for semi-automated tomographic alignment.

3. The manuscript lacks a clear explanation regarding the differences between SR-PB and SR-CB modalities. Specifically, the authors should clarify why SR-PB mode is chosen for large field-of-view imaging and SR-CB for high-resolution imaging. It would be useful to provide a detailed rationale for this choice, especially considering the potential for using a higher magnification objective in SR-PB mode. Could the authors demonstrate whether using a higher magnification objective in SR-PB mode could achieve similar high-resolution imaging as SR-CB mode? Addressing this question would help readers understand the technical rationale and limitations of each modality more thoroughly.

Reply: In the parallel-beam geometry (SR-PB), there is no geometrical magnification and the effective pixel size equals the detector pixel size (0.65 μm with a x10) objective. In the cone-beam geometry (SR-CB), geometric magnification ($M=z_1/z_0$) allows for smaller effective pixel sizes and thus increased resolution at the expense of a smaller field of view.

We have tested higher N.A. objectives, but have not found any advantage in resolution and image quality, e.g. at 20x magnification, which we now explicitly discuss in the *Materials and Methods* section of the revised manuscript (line 163). Note that the point-spread-function (PSF) of the microscope will not improve if the depth of focus becomes smaller than the scintillator or if the signal volume of an X-ray photon is broadened by the energy transfer due to photoelectrons and inelastic interactions of the X-rays in the scintillator, see also the concept of kinetic energy release (KERMA) and linear energy transfer (LET) in radiotherapy. Therefore, even if a higher-resolution microscope (with a higher N.A., oil immersion, etc.) were chosen, the PSF of the whole system would not decrease.

4. The manuscript claims the ability to achieve isotropic resolution, yet I did not find any experiments or results related to resolution measurement or characterization. The paper should include experimental validation of isotropic resolution and provide specific quantitative metrics, such as resolution measurements, in addition to the voxel size.

Reply: We have now included two new figures on the resolution estimates in the Supplementary Information document (Fig. 5, Fig. 6 and Tab. 1 of the revised SI), by fitting a large number of image line-cuts to an erf-profile, each for the SR-CB, SR-PB, and laboratory $\mu\text{-CT}$ cases. The line cuts (e.g. through nuclei) do depend on the steepness of the specific feature, so that they can only provide a lower limit (resolution must be higher than these profile widths). Also, one has to average over many such profiles, which we have now done, see also the revised SI. Importantly, we have no indication that the direction of the profile matters, as expected for isotropic tissues and a correctly implemented CT recording and reconstruction, we therefore can claim isotropy.

5. Do successive rotations of the sample produce motion artifacts during rapid imaging?

Reply: In the parallel-beam configuration (SR-PB) the projections were recorded over a continuous rotation of 360° for a single tomogram. As for the continuous rotation during rapid imaging, no motion artifacts are observed, a detailed technical evaluation can be found in Ref. 26. Typical motion artifacts are neither observed during projection acquisition nor during the reconstruction process.

6. In the supplementary materials, the authors present imaging results of a 3 mm-sized punch biopsy using a stitching algorithm. I am curious whether this method can be applied to larger sample sizes. Specifically, what is the maximum sample size that this approach can support without requiring tissue clearing treatments? Additionally, could the authors discuss whether the sample size might affect the imaging accuracy or resolution? Understanding whether there are any limitations in terms of sample size and how these factors impact the overall imaging performance would provide valuable insight into the scalability and practical applicability of this technique.

Reply: With the SR-PB setup, XPCT has been applied to larger sample sizes, e.g. to 5 mm-sized punch biopsies in (Reichmann et al., PNAS Nexus, 2025), and can be applied up to 8-mm sized punch biopsies in a feasible manner. This is now mentioned in the revised manuscript, where we added the following sentence in the *Summary, Conclusions and Outlook* section: "Punch biopsies of up to 8 mm in diameter can be feasibly scanned with the SR-PB setup, however image quality in terms of resolution and contrast generally decreases with increasing sample size and must be considered." (line 425).

Note that scanning larger sample sizes may increase noise in the image due to higher absorption and scattering of X-ray photons. To increase the penetration depth of X-rays, a higher photon energy is typically chosen. Scalability to large volumes is indeed one of the open challenges in the field. The specimen geometry also matters. For example, the μ -CT scan geometry of laminography may help to cover specimens, which are a few mm in thickness but laterally extended to arbitrary size, by stitching CT scans in the lateral directions.

7. In the imaging results section, there is insufficient comparison with conventional histology based on H&E staining. For example, in Figures 3–5, the authors distinguish different tissue regions and cellular structures using XPCT. However, it would strengthen the manuscript to further compare these results with corresponding H&E staining images to validate and demonstrate the accuracy of the XPCT method. This comparison would provide a more robust assessment of the effectiveness of XPCT for tissue characterization, helping to establish its reliability as a non-invasive alternative to conventional histology.

Reply: We have added microscopic images of H&E-stained slices in Fig. 3 and 4 and have correlated these with virtual slices through the 3D volume obtained by SR-PB and laboratory μ -CT scans, respectively. Correlation of the high-resolution (SR-CB) data set presented in Fig. 5 with H&E histology was already implemented in Fig. 7 for the vessel-like structure and in Fig. 3 of the Supplementary Information document for a complete slice. In particular, we now highlight the differences between endodermal cavities and vessel-like structures in the inner core by correlative imaging in Fig. 3 and 4.

8. The manuscript lacks detailed explanation and comparison regarding the identification and segmentation of features of interest. For instance, in Figure 6c, the purpose and biological significance of the scatter plot are unclear. The authors seem to focus solely on analyzing the accuracy of the segmentation algorithm without elucidating what biological information is being conveyed. Additionally, it is unclear whether the accuracy issues are due to image quality or the segmentation algorithm itself. A comparison of the method's accuracy with traditional HE staining would be beneficial. Furthermore, the authors should emphasize and compare the functionalities and applications that their method enables, which are challenging or unattainable with conventional H&E methods. Providing this information would significantly enhance the understanding of the advantages and potential applications of the XPCT technique in biological research.

Reply: We agree that we used the scatterplot for a rather technical discussion, that it can be used to refine the segmentation. However, we note that the scatterplot also shows effective morphometric parameters obtained by the segmentation, in this case volume and sphericity, and gives information about their distribution, which in turn may be of biological/histological interest, for example when compared to modified states of the myocardial network (e.g. more developed or pathological states).

Regarding the accuracy of the segmentation, we repeat our reply to Reviewer 1:

“The semi-automatic segmentation of the cell nuclei using the Blobfinder algorithm involves a number of steps that are based on the visual (and manual) inspection of the observer, so that the experience of the observer indeed plays a role. First, a mask is manually drawn to include only the nuclei of the ML, while excluding other parts such as the IC or other morphological structures such as endodermal islands. The interpolation of the mask between the drawn sections is then visually inspected. The Blobfinder algorithm itself requires adjustable input parameters, notably "diameter", "probability threshold" and "split sensitivity", which affect the size and number of objects found, and are set manually based on visual inspection. ”

A segmentation process most probably will include false positives and false negatives, and finite pixel size and noise as well as the segmentation process itself additionally can be confounding factors. If segmentation is required for more than illustrative purposes in a study, it may be refined for example using the methods already discussed in the manuscript, i. e. adjusting thresholds to the histograms. While we agree that validation by correlative imaging - e.g. with respect to identifying false positives/negatives - would be valuable and could be emphasized in future studies, we feel that it is beyond the scope of this work. However, we also note that correlation for example with conventional histology is limited, since one would be correlating 3D segmented features to images that are inherently 2D (besides potential distortions after cutting/slicing in H&E histology).

We now state the motivation of this illustrative segmentation more explicitly in the *Results and Discussion* section (line 370), and also revised the *Materials and Methods* section under “Visualization and segmentation” (lines 219-226) to include more details on the segmentation process.

Minor Comments:

1. Lack of scale bars in figures. Many figures lack scale bars, especially the 3D illustrations.

Reply: We thank the reviewer for pointing this out, we have now added scale bars to the 2D sections where they were missing. There are usually no scale bars on the 3D renderings.

2. The rendering of Figure 7.b is not aesthetic, and it seems that only the segmentation results of blood vessels are displayed in 3D, while the original data only shows a single 2D plane. Why is it presented in this manner?

Reply: While the segmented blood vessel can be shown in pseudo-3D, the original data is too dense to be shown in a similar manner. For this reason, we chose the presentation as it is. Of course we always try to find the best visual representation, but for the graphical programs used, we did not find more convincing alternatives here.

Reviewer #3 (Remarks to the Author):

The manuscript focuses on using X-ray phase-contrast tomography (XPCT) for three-dimensional (3D) histology of human heart-forming organoids (HFOs). These organoids are derived from human pluripotent stem cells and model early heart and vasculature development. The authors detail the application of XPCT for visualizing HFOs in their full 3D complexity without destructive slicing, offering high-resolution insights into organoid structure. They also compare synchrotron-based XPCT and laboratory micro-CT (μ CT) for imaging capabilities, segmentation, and visualization. The work demonstrates the utility of XPCT in complementing traditional histology and highlights its potential for high-throughput applications in developmental biology and regenerative medicine.

The manuscript is well-structured, addressing a critical limitation in organoid imaging by presenting XPCT as a non-invasive method for obtaining high-resolution 3D structural data. It integrates innovative imaging techniques with biological applications, advancing both technical and biological fields. The study is significant for organoid-based research, offering a scalable and isotropic approach to tissue visualization. However, the manuscript could benefit from more quantitative validation of imaging data, clearer delineation of the advantages of XPCT over μ CT in practical applications, and a broader discussion of limitations and future improvements.

Reply: We thank the reviewer for his/her positive assessment.

I recommend this paper for publication if it can address the following questions outlined below:

Questions:

1. What specific quantitative metrics, such as contrast-to-noise ratios, spatial resolution, or signal uniformity, can be presented to rigorously compare the performance of XPCT and μ CT in imaging organoid structures?

Reply: In the XPCT-community, several quantitative metrics exist. The most commonly assessed main indicators of image quality are contrast-to-noise ratio (CNR) and resolution. While CNR can be calculated by taking feature/background signal and standard deviation into account, resolution is usually assessed using the half-width-at-half-maximum (HWHM) of line profiles between strongly contrasting features or Fourier-shell-correlation (FSC). We have added two new figures and one table presenting the resolution estimates in the Supplementary Information document (Fig. 5, Fig. 6., and Tab. 1 of the revised SI). These were obtained by fitting line cuts (through edges of nuclei) to an error-function profile for each of the different configurations. We have also added a new figure showing the CNR analysis for each of the different configurations, which can also be found in the Supplementary Information (Fig. 7 of the revised SI). We now discuss the results that are based on the quantitative analysis of the image quality with respect to resolution and CNR throughout the *Results and Discussion* and the *Summary, Conclusions, and Outlook* sections of the revised manuscript (lines 297-301, 339-342, 460-462).

2. How does the proposed XPCT methodology address challenges in scalability, particularly for high-throughput imaging of large datasets or multiple organoid samples, in terms of acquisition speed and data processing?

Reply: In the current implementation of the SR-PB (GINIX) setup, high throughput imaging at beamline P10 is mainly limited by sample changing. To further increase throughput, automated sample handling such as available at beamline P14 (Albers et al. (2024), cited in the manuscript) can be implemented. While acquisition and reconstruction of a dataset in this configuration (2k x 2k x 2k

volume, 0.65 μm pixel size) can be achieved in < 5 minutes, manual sample changing (mounting/dismounting) and semi-automated tomographic alignment will take several more minutes. In the SR-CB configuration, due to a significant decrease in photon flux density (and potentially the requirement of several distance scans in the holographic regime), high-throughput screening by means of rapid data acquisition is currently not possible with data acquisition for a single tomogram taking 2-3 h of time. However, the scan times will be significantly reduced in the future by a higher flux density, more coherent X-rays and improved algorithms (see also our reply below with respect to the 3rd comment) with the coming upgrade to PETRA IV. For in-house measurements (laboratory-based CT), high-throughput screening is feasible as demonstrated in Ref. 48 of the revised manuscript (Reichmann et al., *SciRep*, 2024). As discussed in the manuscript, acquisition time is much larger than for SR-PB scans. The main advantage here is the high accessibility potential of laboratory CT setups.

The (open) challenge to process huge volumes of raw data as a consequence of rapid data acquisition is also currently addressed (e.g. an automated data processing pipeline at the P14, Albers. et al. 2024) and intensively discussed in the X-ray imaging community (a recent review e. g. Zhang et al, *The Innovation*, 2024 on deep learning-empowered data processing pipelines). The implementation of automated data processing pipelines (as well as automated quantitative image analysis as a different topic) is expected in the upcoming upgrades of dedicated X-ray imaging beamlines (also in planning for the upgrade of the GINIX setup as part of the upcoming upgrade to PETRA IV), and for laboratory-based CT scans. This will clearly advance high-throughput capabilities and make XPCT even more accessible to the broad life-science community.

3. What are the fundamental limitations of XPCT in its current implementation, particularly concerning phase retrieval accuracy, resolution trade-offs, or compatibility with live-cell imaging?

With regard to high-resolution XPCT, effective pixel sizes and resolutions below 100 nm have been demonstrated; however, these are still far from the theoretical limit. In the revised manuscript, we have added a paragraph in the *Summary, Conclusions and Outlook* section discussing this topic (line 438), as well as the practical limitations in its current implementation, together with additional references (Ref. 39-43 of the revised MS). As already stated in the manuscript, higher resolution XPCT always comes with a reduction of the FOV and also with extended scanning times (e.g. when compared to the SR-PB setup), which has to be judged in the experimental design. To increase accuracy when solving the phase problem, iterative phase-retrieval approaches and algorithmical advances were recently proposed, with the potential to decrease acquisition time while improving image quality, e.g. Nikitin et al., *Optics Express*, 2024 and Nikitin et al., *Optics Express*, 2025 (now also cited in the revised manuscript, Ref. 42 and 43). Of further note, current limitations will be significantly improved by the upgrade from the synchrotron radiation source PETRA III to PETRA IV at DESY in the coming years, with improved X-ray beam characteristics and optimized beamlines for X-ray imaging.

In general, XPCT is also compatible with live-cell imaging or, in other words, with organoid and other tissue samples in their 'functional environment' since XPCT is not restricted to a specific embedding medium. However, challenges in terms of radiation dose, contrast, and a stable sample environment (to prevent sample movement during data acquisition), for example, must be considered. In the revised manuscript, we have included a general discussion of the use of different embedding media for XPCT in the *Summary, Conclusions and Outlook* section (lines 483-488).

Updated/New Figures:

Updated Figure 3 of the revised manuscript: (f) has been replaced by a comparison between a virtual section through the tomographic reconstruction with an H&E stained histological section. Minor change: Scale bar has been added in the zoom-in in (d).

Updated Figure 4 of the revised manuscript: (a) has been updated by a comparison between a virtual section through the tomographic reconstruction with an H&E stained histological section

Updated Figure 5 of the revised manuscript: a white arrow has been added in (a, yellow zoom-in) which indicates an endothelial cell nucleus of the single-cell endothelial layer that forms the vessel-like structure. Minor changes: Scale bars have been added, where they were missing.

Updated Figure 6 of the revised manuscript: Minor changes - added scale bars.

New Figure 5 of the revised *Supplementary Information* document

New Figure 6 of the revised *Supplementary Information* document

New Figure 7 of the revised *Supplementary Information* document

Reply

(Manuscript #: COMMSBIO-24-4012B)

We are delighted with the positive feedback and would like to thank the reviewers and editors again for the constructive and helpful review process.

Reviewer #1 (Remarks to the Author):

The authors have responded to my comments and clarified all my doubts on the imaging parameters / anatomical structures identification. I am satisfied by the current version of the manuscript.

Reviewer #2 (Remarks to the Author):

I appreciate the authors' responses to my previous comments. The revised manuscript shows clear improvements in both clarity and technical depth.

The additional data and clarifications concerning the imaging methodology and its integration with biological interpretation are helpful and address earlier concerns. In particular, the inclusion of comparative analysis with H&E staining strengthens the validation of the XPCT approach. The expanded description of the computational pipeline and clarification of resolution-related parameters further improve the transparency of the work.

Overall, I find the revisions satisfactory and believe the manuscript is now suitable for publication.

Reviewer #3 (Remarks to the Author):

Thank you for the revision.

I have carefully reviewed the authors' responses and the revised manuscript.

The authors have adequately addressed my previous concerns, and I find the manuscript has improved significantly.

I have no further comments.

I recommend the manuscript for publication in its current form.